

# Controls on leaf water hydrogen and oxygen isotopes: A local
# investigation across seasons and altitude
Jinzhao Liu[a, b*], Huawu Wu[c], Chong Jiang[a], Li Guo[d], Haiwei Zhang[e], Ying Zhao[f]
[a] State Key Laboratory of Loess and Quaternary Geology, Institute of Earth Environment,
Chinese Academy of Sciences, Xi'an 710061, China
[b] National Observation and Research Station of Earth Critical Zone on the Loess Plateau of
Shaanxi, Xi'an, 710061, China
[c] Key Laboratory of Watershed Geographic Sciences, Nanjing Institute of Geography and
Limnology, Chinese Academy of Sciences, Nanjing 210008, China
[d] State Key Laboratory of Hydraulics and Mountain River Engineering & College of Water
Resource and Hydropower, Sichuan University, 610065, Chengdu, China
[e] Institute of Global Environmental Change, Xi'an Jiaotong University, Xi'an, 710054, China
[f] College of resources and environmental engineering, Ludong University, 264025, Yantai,
China
*Corresponding author's email: liujinzhao@ieecas.cn (J. Liu)
## Abstract
The stable oxygen ($\delta^{18}O_{leaf}$) and hydrogen ($\delta^{2}H_{leaf}$) isotopes of leaf water act as a bridge
that connects hydroclimate to plant-derived organic matter. However, it remains unclear



whether the source water (i.e., twig water, soil water, and precipitation) or
meteorological parameters (i.e., temperature, relative humidity, and precipitation) are
the dominant controls on $\delta^{18}O_{leaf}$ and $\delta^2H_{leaf}$. Here, we reported seasonal analysis of
$\delta^{18}O_{leaf}$ and $\delta^2H_{leaf}$ together with isotopes from potential source waters and
meteorological parameters along an elevation transect on the Chinese Loess Plateau.
We found that $\delta^2H_{leaf}$ values were more closely correlated with source water isotopes
than $\delta^{18}O_{leaf}$ values, whereas $\delta^{18}O_{leaf}$ and $\delta^2H_{leaf}$ values were similarly correlated with
meteorological parameters. Dual-isotope analysis showed that the $\delta^{18}O_{leaf}$ and $\delta^2H_{leaf}$
values were closely correlated because of their similar altitudinal and seasonal
responses, and so generated a well-defined isotope line relative to the local meteoric
water line (LMWL). We also compared the measured $\delta^{18}O_{leaf}$ and $\delta^2H_{leaf}$ values with
predicted values by the Craig-Gordon model, and found no significant differences
between them. We demonstrate that the first-order control on $\delta^{18}O_{leaf}$ and $\delta^2H_{leaf}$ values
was the source water, and the second-order control was the enrichment associated with
biochemical and environmental factors.
Short Summary
What controls on leaf water isotopes? We answered the question from two perspectives:
respective and dual isotopes. On the one hand, the $\delta^{18}O$ and $\delta^2H$ values of leaf water
responded to isotopes of potential source water (i.e., twig water, soil water, and
precipitation) and meteorological parameters (i.e., temperature, RH, and precipitation)
differently; On the other hand, dual $\delta^{18}O$ and $\delta^2H$ values of leaf water yielded a





significant regression line, associated with altitude and seasonality.

Keywords: Leaf water, stable isotope, controls, seasonality, altitude

1 Introduction
The stable isotope compositions of oxygen and hydrogen ($\delta^{18}$O and $\delta^2$H, respectively)
are increasingly being used as powerful tracers to follow the path of water from its input
as precipitation, movement through the soil, and ultimately to its release as soil
evaporation and leaf transpiration (Penna and Meerveld, 2019). Leaf water transpiration
plays a key role in regulating water balance at scales ranging from catchment to global.
Terrestrial plants can enrich heavier isotopes ($^2$H and $^{18}$O) in leaf water via evaporative
fractionation through the stoma (Helliker and Ehleinger, 2000; Liu et al., 2015;
Cernusak et al., 2016), which is highly dependent on atmospheric conditions (e.g.,
temperature and relative humidity) and biophysiological processes (Farquhar et al.,
2007; Kahmen et al., 2011; Cernusak et al., 2016). Subsequently, the isotopic signals
from the leaf water are integrated into plant organic matter, such as cellulose (e.g.,
Barbour, 2007; Lehman et al., 2017) and leaf wax (Liu et al., 2016, 2021), as powerful
proxies used for paleoclimate reconstruction (Pagani et al., 2006; Schefuβ et al., 2011;
Hepp et al., 2020). However, although leaf water isotopes are the fundamental
parameters in ecohydrology and organic biosynthesis, we still lack an adequate
understanding of what controls on leaf water isotopes, or the relative importance of
source water and hydroclimates controls leaf water isotopes?






$\delta^{18}O_{leaf}$ and $\delta^2H_{leaf}$ values are influenced firstly by a plant's source water (mainly water
taken up by roots from the soil; Cernusak et al., 2016; Barbour et al., 2017; Munksgaard
et al., 2017), and secondly by the enrichment associated with transpiration (Munksgaard
et al., 2017). Soil water for terrestrial plants generally originates from local precipitation,
and precipitation isotopes vary spatially and temporally, being subject to controls
including temperature, altitude, latitude, distance from the coast, and amount of
precipitation (Bowen, 2010; Bowen and Good, 2015; Cernusak et al., 2016). More
specifically, soil water isotopes are determined by a mixture of individual precipitation
events with distinct isotopic signals and are also affected by evaporation, both of which
lead to the development of isotopic gradients in soil water with depth (Allison et al.,
1983; Liu et al., 2015). A number of studies have shown that the $\delta^{18}O$ and $\delta^2H$ values
of root/xylem water can be used to characterize the water sources used by plants
(Rothfuss and Javaux, 2017; Wu et al., 2018; Wang et al., 2019; Amin et al., 2020; Zhao
et al., 2020; Liu et al., 2021a). These studies rested substantially on the assumption that
no isotopic fractionation of $\delta^{18}O$ and $\delta^2H$ values occurs during water uptake by plant
roots (Dawson and Ehleringer, 1991; Ehleringer and Dawson, 1992; Chen et al., 2020),
except in saline or xeric environments (Lin and Sternberg, 1993; Ellsworth and
Williams, 2007). Some recent studies have shown, however, that the occurrence of
isotopic fractionation during root water uptake was probably more common than
previously thought, especially with respect to $\delta^2H$ values (Zhao et al., 2016; Wang et
al., 2017; Barbeta et al., 2019; Poca et al., 2019; Liu et al., 2021a).






In addition to the plant source water, leaf water is also isotopically enriched through the
evaporative process of transpiration. The enrichment of $^{18}O$ and $^{2}H$ by leaf water
transpiration can be predicted using the Craig-Gordon model (C-G model), which was
originally proposed to describe evaporative enrichment of a freely evaporating water
body (Craig and Gordon, 1965) but has since been modified for plant leaves under
steady-state conditions (Dongmann et al., 1974; Farquhar and Cernusak, 2005).
However, the C-G model fails to explain the intra-leaf heterogeneity of $\delta^{18}O_{leaf}$ and
$\delta^{2}H_{leaf}$ (Cernusak et al., 2016; Liu et al., 2021b), which is currently explained using a
two-pool model (Leaney et al., 1985; Song et al., 2015) and/or an advection diffusion
model, as the *Péclet* effect (Farquhar and Lloyd, 1993; Farquhar and Gan, 2003).
Subsequently, more complicated models have been developed to cover non-steady-state
conditions (Ogée et al., 2007). These models put the emphasis on a mechanistic
understanding of leaf water isotopic fractionation, but the relevant parameters cannot
be strictly constrained or precisely monitored, which hinders the use of these models
under natural conditions (Plavcová et al., 2018).

In this study, we combined the effects of measured source water isotopes and C-G
model-predicted transpiration on $\delta^{18}O_{leaf}$ and $\delta^{2}H_{leaf}$ values. Our objectives were to
deeply understanding the controls on the $\delta^{18}O_{leaf}$ and $\delta^{2}H_{leaf}$ values, and how these
controls vary with the seasons. Based upon these objectives, we repeatedly sampled
soils, twigs, and leaves in May, July, and September (representing spring, summer, and





autumn, respectively) from the same 10 plots that were distributed along an elevation
transect. Simultaneously, we obtained the relevant meteorological parameters (e.g.,
temperature, relative humidity, and precipitation) from sites close to the sampling plots
along the transect and used these to predict the $\delta^{18}O_{leaf}$ and $\delta^2H_{leaf}$ values. The combined
analysis of concurrent measurements of $\delta^{18}O$ and $\delta^2H$ values in soil water, twig water,
and leaf water with the predicted $\delta^{18}O$ and $\delta^2H$ values of leaf water from the C-G model
associated with the surrounding meteorological parameters will help to identify the
factors that control $\delta^{18}O_{leaf}$ and $\delta^2H_{leaf}$ values. Furthermore, we performed an isotope-
based line analysis of the dual $\delta^{18}O$ and $\delta^2H$ values of leaf water, associated with
altitude and seasonality. This study will improve our understanding of the
environmental signals preserved within the $\delta^{18}O$ and $\delta^2H$ values extracted from plant
organic biomarkers associated with leaf water.

## 2. Materials and Methods
### 2.1 Study area
The Qinling Mountains form the dividing line between northern and southern China
and mark the boundary between the watersheds of the Yellow and Yangtze rivers. Mt.
Taibai (Fig. 1; 33. 96 °N, 107.77 °E) rises to 3767 m above sea level (asl) and is the
peak in the Qinling Mountains; it has a warm temperate ecosystem characterized by a
rich diversity of flora and fauna. The mean annual temperature at the bottom of Mt.
Taibai is 12.9°C, and mean annual precipitation is 609.5 mm (Zhang and Liu, 2010).
The climate, soil, and vegetation vary significantly along our slope transect, exhibiting



a remarkable vertical geo-ecological zonation (Fig. 1). The area contains a variety of
climate zones: warm temperate (< 1300 m asl), temperate (1300 - 2600 m asl), cool
temperate (2600 - 3350 m asl), and alpine (> 3350 m asl). The soil types vary from
yellow loess soil at low elevations, spectacular rocky outcrops at middle elevations, and
glacial remnants at high elevations. The vegetation along the transect consists mainly
of coniferous and broadleaf forests and alpine and subalpine vegetation (Fig. 1; Liu,
2021). The dominant species range from *Quercus variabilis*, *Q. aliena*, *Betula*
*albosinensis*, *B. utilis*, *Abies fargessi,* and *Larix chinensis* forests to *Rhododendron*
*clementinae* and *R. concinnum* alpine (Supplementary table S1).
2.2 Sampling strategy
Plants and soils were sampled in May, July, and September 2020, and samples were
collected from 10 plots (3 × 3 m) covering all of the vegetation zones along the
northern slope of Mt. Taibai, extending from 608 to 3533 m asl (Fig. 1). Among the
plots, six (i.e., site 2, 3, 4, 5, 8, 10; Fig. 1) were selected as being the closest to the
weather stations along the elevation transect, and they were used order to obtain the *in-*
*situ* meteorological data for analysis. For the plants, one or two deciduous and
coniferous trees were chosen in each plot, and several large leaves and suberized twigs
were collected for each species. The leaf samples were conducted in the context of the
intact leaves on account of the likely isotopic gradients within a leaf (Helliker and
Ehleringer, 2000; Liu et al., 2016). Our sampling period was between 12 and 15 pm
because maximum diurnal enrichment of the leaf water isotopic composition occurs
during this part of the day (Romero and Feakins, 2011; Liu et al., 2021). The twigs were





collected at the same time by cutting suberized twigs, and all of the twigs were cut into
the samples that were 3-4 cm long. The leaf and twig samples were immediately placed
into glass vials with screw caps and sealed with polyethylene parafilm. For the soils, 3
surface soil samples (less than 10 cm deep) were collected from around the sampled
plants using a small metal scoop at each plot. All sampling plots were located on slopes
far from rivers and surface water bodies, which ensured that the soil water in each plot
was derived exclusively from precipitation. Although the surface soil layers were
collected only as being representative of soil water in this study, these samples could
provide a relatively good source of water for the plants, as supported by a prior study
conducted along the same elevation transect (Zhang and Liu, 2010). The soil samples
were tightly sealed in a polyethylene zipper bag on site. All plant and soil samples were
stored in a cool box (~ 4 °C) in the field and immediately transported to the laboratory.
The altitude of each plot was determined using a handheld GPS unit with an error of ±
5 m.
2.3 Isotope analysis
The water in the plant and soil samples was extracted using an automatic cryogenic
vacuum extraction system (LI-2100 Pro, LICA United Technology Limited, Beijing,
China). The auto-extraction process was set for 3 hours, and the extraction rate of water
from samples was more than 98%. The isotopic composition of soil water was measured
using a Picarro L2130-I isotope water analyzer (Sunnyvale, CA, USA) at the State Key
Laboratory of Loess and Quaternary Geology, Institute of Earth Environment, Chinese
Academy of Sciences. The analytical accuracies were ±0.1‰ for $\delta^{18}O$ and ±1‰ for
$\delta^2$H. The isotopic measurements of twig and leaf water were conducted using an isotope
ratio mass spectrometer coupled to a high-temperature conversion elemental analyzer
(HT2000 EA-IRMS, Delta V Advantage; Thermo Fisher Scientific, Inc. USA) at the
Huake Precision Stable Isotope Laboratory on the campus of Tsinghua Shenzhen
International Graduate School. The measurement precisions were ± 0.2‰ and ± 1‰
for $\delta^{18}$O and $\delta^2$H, respectively. The isotopic composition of $\delta^{18}$O and $\delta^2$H is expressed
as an isotopic ratio:
$\delta_{sample}(‰) = (\frac{R_{sample}-R_{standard}}{R_{standard}}) \times 1000$         (1)
where $\delta_{sample}$ represents $\delta^{18}$O or $\delta^2$H, and $R_{sample}$ and $R_{standard}$ indicate the ratio
of $^{18}$O/$^{16}$O or $^2$H/$^1$H of the sample and standard, respectively. The $\delta^{18}$O and $\delta^2$H values
are reported relative to the Vienna mean standard ocean water (VSMOW). In addition,
the mean monthly $\delta^{18}$O and $\delta^2$H values of precipitation were determined using the
Online Isotope in Precipitation Calculator (Bowen and Revenaugh, 2003).
2.4 Modeling isotopes of leaf water
The C-G equation can be approximated as (Cernusak et al., 2022),
$\delta_e = \delta_s + \varepsilon^+ + \varepsilon_k + (\delta_v - \delta_s - \varepsilon_k) \times \frac{e_a}{e_i}$         (2)
where $\delta_e$ is the predicted $\delta^{18}$O and $\delta^2$H values at the evaporative sites within leaves,
$\delta_s$ is the $\delta^{18}$O and $\delta^2$H values of source water (equivalent to twig water in our study),
$\varepsilon^+$ is the equilibrium fractionation between liquid water and vapour, and $\varepsilon_k$ is the
kinetic fractionation during the diffusion of vapour through the stomata and the
boundary layer.
In our analysis, we calculated $\Delta_v$ (the enrichment of atmospheric vapour relative to





source water) as $\Delta_v = (\delta_v - \delta_s)/(1 + \delta_s)$, and the values of $\Delta_v$ is often close
to $-\varepsilon^+$ at the isotopic steady state (Barbour, 2007; Cernusak et al., 2016); therefore
we can calculate $\delta_v$ as $\delta_v = -\varepsilon^+ + (1 - \varepsilon^+)\delta_s$. In addition, $\frac{e_a}{e_i}$ is the ratio of the
water vapour pressure fraction in the air relative to that in the intercellular spaces and
is equal to the relative humidity (RH) in the air at the steady state (Cernusak et al.,
2022). Thus, Equation (2) can be derived as,
$\delta_e = (1 - h)(\varepsilon^+ + \varepsilon_k) + (1 - \varepsilon^+ h)\delta_s$ (3)
In Equation (3), $\delta_s$ represents the isotopic values of twig water, and $h$ is the mean
annual or monthly RH (MARH or MMRH) in this study. The equilibrium fractionation
($\varepsilon^+$) varies as a function of temperature (Bottinga and Craig, 1969), and can be equated
to $\delta^{18}O$ and $\delta^2H$, as follows (Majoube, 1971):
$\varepsilon_o^+ (‰) = \left[\exp\left(\frac{1.137}{(273+T)^2} \times 10^3 - \frac{0.4156}{273+T} - 2.0667 \times 10^{-3}\right) - 1\right] \times 1000$ (4)
$\varepsilon_H^+ (‰) = \left[\exp\left(\frac{24.844}{(273+T)^2} \times 10^3 - \frac{76.248}{273+T} + 52.612 \times 10^{-3}\right) - 1\right] \times 1000$ (5)
The kinetic fractionation ($\varepsilon_k$) can be calculated for $\delta^{18}O$ and $\delta^2H$ as (Farquhar et al.,

213 2007):

$\varepsilon_k^O (‰) = \frac{28r_s + 19r_b}{r_s + r_b}$ (6)
$\varepsilon_k^H (‰) = \frac{25r_s + 17r_b}{r_s + r_b}$ (7)
where $r_s$ and $r_b$ are the resistances of the stomatal and boundary layers, respectively,
and the inverse of the conductance of the stomatal and boundary layers, respectively.
Previous studies found stomatal and boundary layer conductance values of 0.49 and
2.85 mol m$^{-2}$ s$^{-1}$, respectively (Cernusak et al., 2016; Munksgaard et al., 2017), resulting
in $\varepsilon_k^O$ and $\varepsilon_k^H$ values of 26.7 and 23.8, respectively.



## 2.5 Statistical analysis

Statistical analysis (i.e., the mean, maximum and minimum values, as well as the standard deviation) of the isotopes extracted from the precipitation, soil, twig, and leaf samples was performed to define the range and distribution of the $\delta^{18}O$ and $\delta^2H$ values across the seasons. The Pearson correlation method was used to assess the various correlations between the $\delta^{18}O$ and $\delta^2H$ values among the different water types (i.e., precipitation, soil water, twig water, and leaf water). Hierarchical cluster analysis was used to show the relationships among $\delta^{18}O_{leaf}$ and $\delta^2H_{leaf}$ values and potential source water isotopes ($\delta^{18}O$ and $\delta^2H$ values in precipitation, soil water, twig water, and leaf water), and meteorological parameters such as mean annual and monthly precipitation (MAP and MMP), mean annual and monthly temperature (MAT and MMT), and mean annual and monthly relative humidity (MARH and MMRH). A one-way analysis of variance (ANOVA) combined with a *post hoc* Tukey's least significant difference (LSD) test was performed to identify the significant differences in the isotopic compositions of precipitation, soil, twig, and leaf waters across the months. Comparisons of the relationships of $\delta^{18}O$ and $\delta^2H$ in the soil and leaf water were performed by using analysis of covariance (ANCOVA) to compare slopes across months. The structural equation model (SEM) was used to explain the respective effects of source waters (i.e., twig water, soil water, and precipitation) and meteorological parameters (i.e., temperature, precipitation, and RH) on $\delta^{18}O_{leaf}$ and $\delta^2H_{leaf}$ values. The validated SEMs generated a good model fit, as indicated by a non-signi*fi*cant $\chi^2$ test ($p > 0.05$), a high comparative fit index (CFI > 0.95), and a low root mean square error of approximation


(RMSEA < 0.05). A special SEM was constructed based on the Mantel R values in
AMOS (version 24.0.0). Moreover, we used the Hybrid Single-Particle Lagrangian
Integrated Trajectory (HYSPLIT) model (Draxler and Rolph, 2003) to calculate air
mass back-trajectory for a central site (34.13°N, 107.83°E, 2270 m asl) in the study
area. These trajectories were initiated four times daily (at 00:00, 06:00, 12:00, and 18:00
UTC) and their air parcel was released at 2300 m asl for May, July, and September 2020
and moved backwards by winds for 120 h (5 days).

3. Results
3.1 Differing response of $\delta^{18}$O and $\delta^2$H values of leaf water
The measured $\delta^{18}$O and $\delta^2$H values of leaf water responded differently to source water
isotopes (Fig. 2a) and meteorological parameters (Fig. 2b) across the seasons. Cluster
analysis showed that the leaf water $\delta^{18}$O and $\delta^2$H values ($\delta^{18}$O$_{leaf}$ and $\delta^2$H$_{leaf}$) were
clustered with the twig water $\delta^{18}$O and $\delta^2$H values ($\delta^{18}$O$_{twig}$ and $\delta^2$H$_{twig}$; Fig. 2a), and
also with MARH, MAT, and MMT (Fig. 2b). The $\delta^2$H$_{leaf}$ values were more closely
correlated with isotopes of the potential source waters (e.g., twig water, soil water, and
precipitation) than the $\delta^{18}$O$_{leaf}$ values in different months (Fig. 2a), whereas leaf water
$\delta^{18}$O and $\delta^2$H values were comparatively correlated with meteorological parameters
(Fig. 2b) across months. These correlations were more significant in summer (July) and
autumn (September) than those in spring (May).

3.2 Comparisons of measured and predicted $\delta^{18}$O and $\delta^2$H values of leaf water





The $\delta^{18}O_{leaf}$ and $\delta^{2}H_{leaf}$ values predicted by the C-G model were compared with the
measured $\delta^{18}O$ and $\delta^{2}H$ values across all three months (Fig. 3). The C-G models
explained 49% and 70% of the observed variations in the $\delta^{18}O_{leaf}$ and $\delta^{2}H_{leaf}$ values,
respectively (Fig. 3a, c). The slopes of the relationships for both $\delta^{18}O$ and $\delta^{2}H$ values
of leaf water were less than one, which suggests that part of the bulk leaf water is
derived from unenriched vein water. However, there were no significant differences in
$\delta^{18}O_{leaf}$ ($p = 0.54$; Fig. 3b) and $\delta^{2}H_{leaf}$ values ($p = 0.93$; Fig. 3d) between the C-G model
predicted values and the measured values.

3.3 Variation of $\delta^{18}O$ and $\delta^{2}H$ values of different waters with seasons and altitude
There was a significant correlation between $\delta^{18}O_{leaf}$ and $\delta^{2}H_{leaf}$ values ($R^2 = 0.81$, $p <$
0.01; Fig. 4), with significant clusters of $\delta^{18}O_{leaf}$ and $\delta^{2}H_{leaf}$ values across the months,
and values being higher in May, intermediate in July, and lower in September (Fig. 4).
Within each month, the $\delta^{18}O_{leaf}$ and $\delta^{2}H_{leaf}$ values were depleted in $^{2}H$ and $^{18}O$ at higher
altitudes relative to lower altitudes. Likewise, the potential types of source water (i.e.,
twig water, soil water, and precipitation) exhibited consistent variations across the
months, showing values that were relatively higher in May, intermediate in July, and
lower in September (Supplementary Fig. S1). The correlations between $\delta^{18}O$ and $\delta^{2}H$
values among the source waters were also significant (Supplementary Fig. S2), but the
slopes and coefficients of determination ($R^2$) between the $\delta^{18}O$ and $\delta^{2}H$ values showed
decreasing trends for precipitation, soil water, twig water, and leaf water from the three
sampling months, except for soil water in May (Supplementary Fig. S2). In addition,



the ANCOVA tests showed no significant differences for the regression lines for
precipitation (df = 0.47, $F$ = 2.49, $p$ = 0.11 > 0.05), twig water (df = 53.2, $F$ = 0.42, $p$ =
0.66 > 0.05), and leaf water (df = 437.3, $F$ = 2.78, $p$ = 0.08 > 0.05) across the months,
but a significant difference for soil water across the months (df = 308.8, $F$ = 10.9, $p$ <

291 0.05).


4. Discussion
4.1 $\delta^{18}O$ and $\delta^{2}H$ values of leaf water
A recent global meta-analysis indicated that $\delta^{18}O_{leaf}$ and $\delta^{2}H_{leaf}$ values reflect
environmental drivers differently and showed that $\delta^{2}H_{leaf}$ values more strongly reflect
xylem water and atmospheric vapour $\delta^{2}H$ values, whereas $\delta^{18}O_{leaf}$ values more strongly
reflect air relative humidity (Cernusak et al., 2022). Our seasonal and localized
observations along an elevation transect on the Chinese Loess Plateau supported these
differing responses of $\delta^{18}O_{leaf}$ and $\delta^{2}H_{leaf}$ values to isotopic composition of the potential
source water and meteorological parameters (Fig. 2). We found stronger correlations
between $\delta^{2}H_{leaf}$ and isotope values of the source water (twig water, soil water, and
precipitation) than between $\delta^{18}O_{leaf}$ values and the source water isotope values (Fig. 2a).
This is consistent with the global meta-analysis (Cernusak et al., 2022). However, our
localized observational study did not show a significantly different response of $\delta^{18}O_{leaf}$
and $\delta^{2}H_{leaf}$ values to meteorological parameters, and they responded at an almost
equivalent magnitude (Fig. 2b). These observations suggest that plant organic isotopic
proxies such as leaf wax (Sachse et al., 2012; Liu et al., 2016) and cellulose (Barbour,



2007; Lehman et al., 2017), which originate from $\delta^{18}O_{leaf}$ and $\delta^2H_{leaf}$ values, can provide
comparative information that indicates climatic signals (e.g., temperature, RH, and
precipitation) in natural archives. These results argued with the recent global meta-
analysis that $\delta^{18}O_{leaf}$ and $\delta^2H_{leaf}$ values reflect climatic parameters (i.e., RH and
temperature) differently (Cernusak et al., 2022).

The results of the cluster analysis showed that the isotope values of leaf water ($\delta^{18}O_{leaf}$
and $\delta^2H_{leaf}$) and twig water ($\delta^{18}O_{twig}$ and $\delta^2H_{twig}$) were clustered into one group, but
those of soil water ($\delta^{18}O_{soil}$ and $\delta^2H_{soil}$) and precipitation ($\delta^{18}O_p$ and $\delta^2H_p$) were
clustered into another (Fig. 2a). This indicates that the direct source water of $\delta^{18}O_{leaf}$
and $\delta^2H_{leaf}$ should be $\delta^{18}O_{twig}$ and $\delta^2H_{twig}$, providing the source water isotope basis for
the C-G model. In the C-G model (see Equation 2), besides the source water isotopes,
the equilibrium fractionation factor ($\varepsilon^+$) and atmospheric vapour enrichment ($\Delta_v$)
depend on the temperature at the isotopic steady state. Thus, the $\delta^{18}O_{leaf}$ and $\delta^2H_{leaf}$
values were predicted to be associated primarily with temperature, RH, and source
water, which is consistent with the results from the cluster analysis that the $\delta^{18}O_{leaf}$ and
$\delta^2H_{leaf}$ values were clustered with temperature (MAT and MMT) and RH (MARH; Fig.
2b). Based on the C-G model, we plotted the measured and predicted $\delta^{18}O_{leaf}$ and $\delta^2H_{leaf}$
values (Fig. 3a, c) and observed no significant differences between the measured and
predicted values of $\delta^{18}O_{leaf}$ and $\delta^2H_{leaf}$ values (Fig. 3b, d). Although the slopes of the
predicted and measured $\delta^{18}O_{leaf}$ and $\delta^2H_{leaf}$ values were less than one, the C-G model
still provides a reasonable framework for guiding analysis of the different controls on



$\delta^{18}O_{leaf}$ and $\delta^2H_{leaf}$ values.

**4.2 Dual $\delta^{18}O$ and $\delta^2H$ plots of leaf water**
There was a significant linear correlation between the $\delta^{18}O_{leaf}$ and $\delta^2H_{leaf}$ values, with
remarkable clusters associated with the three months analyzed in this study (Fig. 4). As
is well-known, the LMWL, generated by precipitation $\delta^{18}O$ and $\delta^2H$ values at the local
scale, serves as an important reference line for inter-comparisons among different
waters. Furthermore, the regression lines of the $\delta^{18}O$ and $\delta^2H$ values from soil water,
twig water, and leaf water (Supplementary Fig. S2) suggest that the leaf water isotopes
could well inherit isotopic signals of source waters that originate from twig water, soil
water, and ultimately precipitation. The slopes and intercepts of the $\delta^{18}O$ and $\delta^2H$ values
decreased significantly from precipitation, soil water, twig water, and leaf water for
each month, except for soil water in May (Supplementary Fig. S2). Such patterns have
been observed in the a number of previous calibration studies (Brooks et al., 2010;
Evaristo et al., 2015; Sprenger et al., 2016, 2017; Wang et al., 2017; Benettin et al.,
2018; Barbeta et al., 2019; Penna and Meerveld, 2019; Liu et al., 2021a). The slopes of
the LMWLs were lower in July (6.79) relative to those from May (7.04) and September
(6.85), but were not significantly different (ANCOVA test: df = 0.47, $F$ = 2.49, $p$ = 0.11 >
0.05). This suggests that the local water vapour from precipitation was derived from the
same source across the seasons, but was subject to different intensities of evaporation
as the temperature changed through the seasons (Li et al., 2019; Wu et al., 2019, 2021).
The slopes of the $\delta^{18}O$ and $\delta^2H$ values from the soil, twig, and leaf waters were also



much smaller than the LMWLs across the months due to the occurrence of secondary
evaporation in the other water types.

In the dual isotope plot of leaf water, there were well-defined clusters of $\delta^{18}O_{leaf}$ and
$\delta^2H_{leaf}$ values across the three months: $^{18}O$ and $^2H$ were depleted in September, there
were intermediate values in July, and $^{18}O$ and $^2H$ were enriched in May (Fig. 4). When
focusing on each month, relatively higher isotopic values occurred at low elevations,
but lower isotopic values were present at high elevations despite there being no, or only
weak, correlations between the the $\delta^{18}O_{leaf}$ and $\delta^2H_{leaf}$ values and altitude
(Supplementary Fig. S3). The correlations between the $\delta^{18}O_{leaf}$ and $\delta^2H_{leaf}$ values and
altitude, and between the $\delta^{18}O_{twig}$ and $\delta^2H_{twig}$ values and altitude, were not significant
and weak across the three months; however, the $\delta^{18}O_p$ and $\delta^2H_p$, and also the $\delta^{18}O_{soil}$
and $\delta^2H_{soil}$ values, were significantly correlated with altitude (Supplementary Fig. S3),
which suggests that besides source water (precipitation and soil water), other factors
associated with plants also affect the $\delta^{18}O_{leaf}$ and $\delta^2H_{leaf}$ values.

The dual isotope plot of $\delta^{18}O_{leaf}$ and $\delta^2H_{leaf}$ values show a significant isotope line: $y =$
$4.52x - 50.7$ ($R^2 = 0.81$, $p < 0.01$; Fig. 4), but relatively shallower slopes (3.53, 1.86,
and 2.81 in May, July, and September, respectively) of $\delta^{18}O_{leaf}$ and $\delta^2H_{leaf}$ values were
observed across the seasons (Supplementary Fig. S2). Such a correlation was supported
by a recent study that conducted consecutive measurements of $\delta^{18}O$ and $\delta^2H$ values in
xylem/leaf water in Switzerland and indicated that leaf water provided great potential

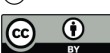



to determine the source water of plants (Benettin et al., 2021). Our local study showed
remarkable clusters in the measured (Fig. 4) and the C-G model predicted (Fig. 3)
$\delta^{18}O_{leaf}$ and $\delta^2H_{leaf}$ values across the months and the consistencies of respective $\delta^{18}O_{leaf}$
and $\delta^2H_{leaf}$ values with potential source water isotopes across months (Supplementary
Fig. S1). These findings of temporally consistent dynamics among the water types (i.e.,
precipitation, soil water, twig/stem water, and leaf water) have been observed in a
number of previous studies (Phillips and Ehleringer, 1995; Cernusak et al., 2005;
Sprenger et al., 2016; Berry et al., 2017; Liu et al., 2021a).

The isotopic inheritance from precipitation to leaf water indicate that seasonal
variations of $\delta^{18}O_p$ and $\delta^2H_p$ values are the first-order control on the temporal patterns
seen in the leaf water. The seasonal dynamics of the $\delta^{18}O_p$ and $\delta^2H_p$ values reflect the
combined effects of such things as temperature, altitude, and precipitation amount,
which are associated with orographic conditions, as well as sub-cloud evaporation,
moisture recycling, and differences in the vapor source (Dansgaard, 1964; McGuire and
McDonnell, 2007; Li et al., 2016; Penna and Meerveld, 2019; Wu et al., 2019). In this
study, we used the HYSPLIT model to demonstrate the ultimate cause of the seasonal
variations of $\delta^{18}O_{leaf}$ and $\delta^2H_{leaf}$ values; that is, the monthly dynamics of the $\delta^{18}O_p$ and
$\delta^2H_p$ values. The monthly variations of the $\delta^{18}O_p$ and $\delta^2H_p$ values from the Global
Network for Isotopes in Precipitation (GNIP, http://www.iaea.org/) at Xi'an station
(1985-1992 AD), which is ~100 km from our study transect, were enriched in $^{18}O$ and
$^2H$ in May relative to July and September (Fig. 5a, b). The cluster mean of the moisture



transport routes from HYSPLIT (Draxler and Rolph, 2003) and the climatological 850
hPa wind vectors showed that the main moisture sources were from western China and
central Asia in May, the China-India Peninsula and Bay of Bengal, and local moisture
recycling and convection (Fig. 5c, d, e). The seasonal variations in $\delta^{18}O_p$ and $\delta^2H_p$
values are consistently related to the onset, advancement, and retreat of the Asian
summer monsoon and associated changes in the large-scale monsoon circulation (e.g.,
Zhang et al., 2020, 2021). As the summer monsoon starts in mid-May, the rainfall
season starts in southern China; however, our study area is controlled mainly by
moisture from the westerlies (Chiang et al., 2015) with relatively higher vapour, $\delta^{18}O_p$,
and $\delta^2H_p$ values (Fig. 5c, a, b). In July, the summer monsoon reaches its strongest phase
and the rainfall belt shifts to central and northern China, where the southerly wind
brings plenty of moisture from the China-India Peninsula and the Bay of Bengal with
lower vapour, $\delta^{18}O_p$, and $\delta^2H_p$ values (Fig. 5d, a, b). When the summer monsoon
withdraws in September, the study area is controlled mainly by moisture from local
moisture recycling and convection (Fig. 5e). Soil water stores the June-August
monsoon rainfall with its lower $\delta^{18}O$ and $\delta^2H$ values, resulting in even lower $\delta^{18}O_p$ and
$\delta^2H_p$ values in September than in July (Supplementary Fig. S1), and thus resulting in
significantly lower $\delta^{18}O$ and $\delta^2H$ values of leaf water (Fig. 4).

4.3 Framework of controls for $\delta^{18}O$ and $\delta^2H$ values of leaf water
To delineate the mechanisms that control the $\delta^{18}O_{leaf}$ and $\delta^2H_{leaf}$ values, we used the
SEMs to quantify the complex interactions among $\delta^{18}O_{leaf}$ or $\delta^2H_{leaf}$ values, and source





waters, and meteorological parameters (Fig. 6). The coefficients of determination ($R^2$)
were 0.48 and 0.71 for the $\delta^{18}O_{leaf}$ and $\delta^2H_{leaf}$ values, respectively, indicating that the
models explained more variance for $\delta^2H_{leaf}$ values than $\delta^{18}O_{leaf}$ values (Fig. 6). The
SEMs showed that potential source waters (i.e., twig water, soil water, and precipitation)
had stronger effects on $\delta^2H_{leaf}$ relative to $\delta^{18}O_{leaf}$ values, while the meteorological
parameters showed weak effects on both $\delta^{18}O_{leaf}$ and $\delta^2H_{leaf}$ values (a little larger for
$\delta^2H_{leaf}$ than $\delta^{18}O_{leaf}$ values). This is consistent with our above correlation analysis (Fig.
2). Surprisingly, MMT had significant and strong effects on $\delta^{18}O_p$ and $\delta^2H_p$ values,
suggesting that temperature plays a key role in determining $\delta^{18}O_p$ and $\delta^2H_p$ values, but
this finding is not discussed further here. Collectively, the SEMs also showed that
source water exerts the first-order control but affects $\delta^{18}O_{leaf}$ and $\delta^2H_{leaf}$ differently; the
meteorological parameters had a weak control on $\delta^{18}O_{leaf}$ and $\delta^2H_{leaf}$, with a relatively
stronger effect on $\delta^2H_{leaf}$ than $\delta^{18}O_{leaf}$ values.

A schematic representation of the controls on $\delta^{18}O_{leaf}$ and $\delta^2H_{leaf}$ values (respective and
dual) is shown in Fig. 7, and involves multiple processes associated with the
hydroclimatic and biochemical factors that affect $\delta^{18}O_{leaf}$ and $\delta^2H_{leaf}$ values. The
meteorological parameters (temperature, RH, and precipitation) exerted distinct effects
on the $\delta^{18}O$ and $\delta^2H$ values of the source water, and thus on the $\delta^{18}O_{leaf}$ and $\delta^2H_{leaf}$ values,
as demonstrated above by the SEM. Significant isotopic fractionation occurred mainly
at two key locations across the vertical soil profiles and leaf architectures from
precipitation to leaf water. First, an isotopic gradient across the vertical soil profile





appeared because of evaporation from the surface soil layers (Ehleringer et al., 1992;
Goldsmith et al., 2012; Evaristo et al., 2015). This evaporative isotopic fractionation
causes an isotopic linear trajectory down soil profile (Goldsmith et al., 2012; Rothfuss
and Javaux, 2017; Wu et al., 2018; Wang et al., 2019; Amin et al., 2020; Zhao et al.,
2020; Liu et al., 2021a). Second, there were significant isotopic heterogeneities
associated with the $\delta^{18}O_{leaf}$ (Helliker and Ehleringer, 2000; Farquhar and Gan, 2003;
Gan et al., 2003; Song et al., 2015) and $\delta^2H_{leaf}$ values (Šantrůček et al., 2007; Liu et al.,
2016; Liu et al., 2021b) within a leaf, which depends substantially on veinal structures
(Liu et al., 2021b). The within-leaf heterogeneity of the $\delta^{18}O_{leaf}$ and $\delta^2H_{leaf}$ values can
be explained using the *Péclet*-modified C-G model (Gan et al., 2003; Farquhar and Gan,
2003; Cernusak et al., 2005, 2016).

Moreover, the hydroclimatic factors (e.g., temperature, RH, precipitation, etc.) varied
with altitude and seasonality, yielding an isotopic water line (LWL) in the dual-isotope
plot (Fig. 4). The slope of the LWL was shallower than the LMWL, with an intersection
angle $\theta$ (Fig. 7). We speculate that $\theta$ probably varies with the hydroclimatic and
biochemical factors associated with evaporation, transpiration, and biochemistry, but
the relationship between $\theta$ and these hydroclimatic and biochemical factors required
further exploration. Overall, the LWL is controlled primarily by altitude and seasonality,
as these are the main influences on the hydroclimatic and biochemical factors.

5 Conclusion





Along an elevation transect on the Chinese Loess Plateau, precipitation, soil water, twig
water, and leaf water were repeatedly sampled to explore the controls on $\delta^{18}O_{leaf}$ and
$\delta^2H_{leaf}$ values associated with meteorological parameters and source water. The effects
of meteorological parameters and source water on $\delta^{18}O_{leaf}$ and $\delta^2H_{leaf}$ values were
different, and the dual $\delta^{18}O_{leaf}$ and $\delta^2H_{leaf}$ plot generated an isotopic line. The $\delta^{18}O_{leaf}$
and $\delta^2H_{leaf}$ values were controlled by the combined effects of source water and
hydroclimate that varied with altitude and season.

**Competing interests**
The authors declare that they have no known competing financial interests or personal
relationships that could have appeared to influence the work reported in this paper.

**Acknowledgement**
We thank X. Cao and M. Xing for help with laboratory assistance, and Y. Cheng for the
help in the field. We thank Profs. J. J. McDonnell and L. A. Cernusak for discussing
and editing the paper. This work was supported by the Chinese Academy of Sciences
(XDB40000000; XAB2019B02; ZDBS-LY-DQC033; 132B61KYSB20170005) and
National Natural Science Foundation of China (42073017).

**Author contribution**
J.L. conceived the idea of research, and performed the data analysis. J.L., H.W., and
H.Z. wrote the manuscript. L.G. and Y.Z. edited the paper. J.L. and C.J. performed the



lab work. All authors contributed to discuss the results.

**Data availability statement**

Data related to this article can be found in Electric Annex and Mendeley Data
(https://data.mendeley.com/drafts/t44wybgpr3).

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

Contributions of local terrestrial evaporation and transpiration to precipitation using
$\delta^{18}O$ and D-excess as a proxy in Shiyang inland river basin in China, Global Planet.
Change, 146, 140–151, 2016.
Li, Z., Li, Z., Yu, H., Song, L., and Ma, J.: Environmental significance and zonal
characteristics of stable isotope of atmospheric precipitation in arid Central Asia. Atmos.
Res., 227, 24–40, 2019.
Lin, G. H., and Sternberg, L. S. L.: Hydrogen isotopic fractionation by plant roots
during water uptake in coastal wetland plants. Stable Isotopic and Plant Carbon/Water
Relations, Academic Press, New York, pp. 497–510, 1993.
Liu, J., Liu, W., and An, Z.: Insight into the reasons of leaf wax $\delta D_{n-alkane}$ values between
grasses and woods, Sci. Bull., 60, 549–555, 2015.
Liu, J., Liu, W., An, Z., and Yang, H.: Different hydrogen isotope fractionations during
lipid formation in higher plants: Implications for paleohydrology, Sci. Report, 6, 19711,

611    2016.

Liu, J., Wu, H., Cheng, Y., Jin, Z., and Hu, J.: Stable isotope analysis of soil and plant
water in a pair of natural grassland and understory of planted forestland on the Chinese
Loess Plateau, Agr. Water Manage., 249, 106800, 2021a.
Liu, J., An, Z., and Lin, G.: Intra-leaf heterogeneities of hydrogen isotope compositions
in leaf water and leaf wax of monocots and dicots, Sci. Total Environ., 770, 145258,





2021b.
Liu, J.: Seasonality of the altitude effect on leaf wax n-alkane distributions, hydrogen
and carbon isotopes along an arid transect in the Qinling Mountains. Sci. Total Environ.,

620     778, 146272, 2021.

Majoube M. Fractionnement en oxygen-18 et en deuterium entre l'eau et sa vapeur.
Journal de Chimie et Physique 68, 1423–1436, 1971.
McGuire, K., and McDonnell J. J.: Stable isotope tracers in watershed hydrology, in
Stable Isotopes in Ecology and Environmental Science, Ecological Methods and
Concepts Series, pp. 334–374, 2007.
Munksgaard, N. C., Cheesman, A. W., English, N. B., Zwart, C., Kahmen, A., and
Cernusak, L. A.: Identifying drivers of leaf water and cellulose stable isotope
enrichment in Eucalyptus in northern Australia, Oecologia, 183, 31–43, 2017.
Ogée, J., Cuntz, M., Peylin, P., Bariac, T., 2007. Non-steady-state, non-uniform
transpiration rate and leaf anatomy effects on the progressive stable isotope enrichment
of leaf water along monocot leaves. Plant Cell Environ. 30, 367–387.
Pagani, M., Pedentchouk, N., Huber, M., Sluijs, A., Schouten, S., Brinkhuis, H., Damsté,
J. S. S., and Dichens, G. R.: Arctic hydrology during global warming at the
Palaeocene/Eocene thermal maximum, Nature, 442, 671–675, 2006.
Penna, D., and van Meerveld, H. J.: Spatial variability in the isotopic composition of
water in small catchments and its effect on hydrograph separation, WIREs Water, e1367,

637     2019.

Phillips, S. L., and Ehleringer, J. R.: Limited uptake of summer precipitation by big





tooth maple (*Acer grandidentatum* Nutt) and Gambels oak (*Quercus gambelii* Nutt),
Trees, 9, 214–219, 1995.
Plavcová, L., Hronková, M., Šimková, M., Květoň, J., Vráblová, M., Kubásek, J.,
Šantrůček, J.: Seasonal variation of $\delta^{18}$O and $\delta^2$H in leaf water of *Fagus sylvatica* L.
and related water compartments, J. Plant Physiol., 227, 56–65, 2018.
Poca, M., Coomans, O., Urcelay, C., Zeballos, S. R., Bodé, S., and Boecks, P.: Isotope
fractionation during root water uptake by *Acacia caven* is enhanced by arbuscular
mycorrhizas, Plant Soil, 441, 485–497, 2019.
Romero, I.C., Feakins, S.I., 2011. Spatial gradients in plant leaf wax D/H across a
coastal salt marsh in southern California. Org. Geochem. 42, 618–629.
Rothfuss, Y., and Javaux, M.: Reviews and syntheses: isotopic approaches to quantify
root water uptake: a review and comparison of methods, Biogeosciences, 14, 2199–

651    2224, 2017.

Sachse, D., Billault, I., Bowen, G.J., Chikaraishi, Y., Dawson, T.E., Feakins, S.J.,
Freeman, K.H., Magill, C.R., McInerney, F.A., van der Meer, M.T.J., Polissar, P.J.,
Robins, R.J., Sachs, J.P., Schmidt, H.L., Sessions, A.L., White, J.W.C., West, J.B.,
Kahmen, A., 2012. Molecular paleoyhydrology: interpreting the hydrogen-isotopic
composition of lipid biomarkers from photosynthesizing organisms. Annu. Rev. Earth
Planet. Sci. 40, 221–249.
Šantrůček, J., Květoň, J., Šetlík, J., Bulíčková, L., 2007. Spatial variation of deuterium
enrichment in bulk water of snowgun leaves. Plant Physiol. 143, 88–97.
Song, X., Loucos, K. E., Simonin, K. A., Farquhar, G. D., and Barbour, M. M.:



Measurements of transpiration isotopologues and leaf water to assess enrichment
models in cotton, New Phytol., 206, 637–646, 2015.
Schefuβ, E., Kuhlmann, H., Mollenhauer, G., Prange, M., and Pätzold, J.: Forcing of
wet phases in Southeast Africa over the past 17,000 year, Nature, 480, 22–29, 2011.
Sprenger, M., Leistert, H., Gimbel, K., and Weiler, M.: Illuminating hydrological
processes at the soil-vegetation-atmosphere interface with water stable isotopes, Rev.
Geophys., 54, 674–704, 2016.
Sprenger, M., Tetzlaff, D., and Soulsby, S.: Soil water stable isotopes reveal evaporation
dynamics at the soil-plant-atmosphere interface of the critical zone, Hydrol. Earth Syst.
Sci., 21, 3839–3858, 2017.
Wang, J., Fu, B., Lu, N., and Zhang, L.: Seasonal variation in water uptake patterns of
three plant species based on stable isotopes in the semi-arid Loess Plateau, Sci. Total
Environ., 609, 27–37, 2017.
Wang, J., Lu, N., and Fu, B.: Inter-comparison of stable isotope mixing models for
determining plant water source partitioning, Sci. Total Environ. 666, 685–693, 2019b.
Wu, H., Li, J., Li, X., He, B., Liu, J., Jiang, Z., and Zhang, C.: Contrasting response of
coexisting plant′s water-use patterns to experimental precipitation manipulation in an
alpine grassland community of Qinghai Lake watershed, China, PLoS One, 13,
e0194242, 2018.
Wu, H., Wu, J., Sakiev, K., Liu, J., Li, J., He, B., Liu, Y., and Shen, B.: Spatial and
temporal variability of stable isotopes ($\delta^{18}$O and $\delta^2$H) in surface waters of arid,
mountainous Central Asia, Hydrol. Process. 33, 1658–1669, 2019.



Wu, H., Huang, Q., Fu, C., Song, F., Liu, J., Li, J.: Stable isotope signatures of river
and lake water from Poyang Lake, China: Implications for river-lake interactions. J.
Hydrol. 592, 125619, 2021.
Zhang, P., and Liu, W.: Effect of plant life form on relationship between δD values of
leaf wax *n*-alkanes and altitude along Mount Taibai, China, Org. Geochem., 42, 100–

688    107, 2010.

Zhao, L., Wang, L., Cernusak, L. A., Liu, X., Xiao, H., Zhou, M., and Zhang, S.:
Significant difference in hydrogen isotope composition between xylem and tissue water
in *Populus Euphratica*, Plant Cell Environ., 39, 1848–1857, 2016.
Zhao, Y., Wang, Y., He, M., Tong, Y., Zhou, J., Guo, X., Liu, J., Zhang, X.: Transference
of *Robinia pseudoacacia* water-use patterns from deep to shallow soil layers during the
transition period between the dry and rainy seasons in a waterlimited region, For. Ecol.
Manag., 457, 117727, 2020.
Zhang, H., Cheng, H., Cai, Y., Spötl, C., Sinha, A., Kathayat, G., Li, H.: Effect of
precipitation seasonality on annual oxygen isotopic composition in the area of spring
persistent rain in southeastern China and its paleoclimatic implication, Clim. Past, 16,
211–225, 2020.
Zhang, H., Zhang, X., Cai, Y., Sinha, A., Spötl, C., Baker, J., Kathayat, G., Liu, Z., Tian,
Y., and Lu, J.: A data-model comparison pinpoints Holocene spatiotemporal pattern of
East Asian summer monsoon, Quat. Sci. Rev., 261, 106911, 2021.




**Figure captions**

**Fig. 1** Sample sites (red dots) and weather stations (open triangles) that distribute along vertical vegetation zones across the Mt. Taibai transect on the Chinese Loess Plateau (a). The meteorological parameters (precipitation, temperature, and RH) vary with stations along elevation transect (b). Mean annul (MAP, MAT, MARH) and montly (MMP, MMT, MMRH) precipitation, temperature, and relative humidity. The subscripts refer to the month. The vertical vegetation distribution was adopted from Liu, 2021.

**Fig. 2** Heatmaps of correlations ($r$) between leaf water $\delta^{18}O$ and $\delta^{2}H$ values and potential source water $\delta^{18}O$ and $\delta^{2}H$ values (twig water, soil water, and precipitation $\delta^{18}O$ and $\delta^{2}H$ values; a), and meteorological parameters (e.g., MAP, MMP, MAT, MMT, MARH, MMRH). The hierarchical cluster analysis of the isotopes of leaf water and source water (a), and meteorologica parameters (b). The subscripts (p, soil, twig, leaf) refer to precipitation, soil water, twig water, and leaf water. * Corrected significance at $p < 0.05$; ** corrected significance at $p < 0.01$; *** corrected significance at $p < 0.001$.

**Fig. 3** Measured leaf water isotopic composition for $\delta^{18}O$ (a) and $\delta^{2}H$ (c) values against values predicted by the C-G model. Boxplots show no significant differences for $\delta^{18}O$ (b) and $\delta^{2}H$ (d) values between measured and predicted leaf water. The dotted lines show one-to-one lines.

**Fig. 4** Correlation of leaf water $\delta^{18}O$ and $\delta^{2}H$ values across months and altitude. Leaf water $\delta^{18}O$ and $\delta^{2}H$ values were the higher in May, intermediate in July, and lower in September, and while within each month, those isotopic values were relatively lower at high altitudes and higher in lower altitudes.

**Fig. 5** Variation of monthly mean precipitation $\delta^{18}O$ (a) and $\delta^{2}H$ (b) values at Xi'an station from Global Network of Isotopes in Precipitation (GNIP) and cluster mean of moisture transport routes using HYSPLIT model in May (c), July (d) and September (e), 2020. Background in (c-e) is the average precipitation (mm/day) and 850 hPa wind vectors (arrows, m/s) in May (c), July (d) and September (e) in 1979-2016 AD based on the database of the Global Precipitation Climatology Center (GPCC) (Becker et al.,

2011) and the Modern-Era Retrospective analysis for Research and Applications
(Rienecker et al., 2011).
**Fig. 6** Structural equation model (SEM) of leaf water $\delta^{18}O$ (a) and $\delta^2H$ (b) values. The
structural equation models considered all plausible pathways. Solid lines indicate
significant positive (red) or negative (blue) effects, and dashed lines indicate non-
significant effects. Grey lines indicate correlations between two variables. Numbers on
the arrow indicate significant standardized path coefficients, proportional to the arrow
width. The coefficients of determination ($R^2$) represent the proportion of variance
explained by the model.
**Fig. 7** Schematics of the respective and dual isotopes of $\delta^{18}O$ and $\delta^2H$ values from
precipitation to leaf water, associated with physical (evaporation at soil profile and
transpiration at leaf level) and biochemical processes. The dual isotopes of $\delta^{18}O$ and
$\delta^2H$ values yield an isotopic water line, the slope of which was lower than the LMWL.
The intersected angle varied with hydroclimates, associated with altitude and
seasonality.


















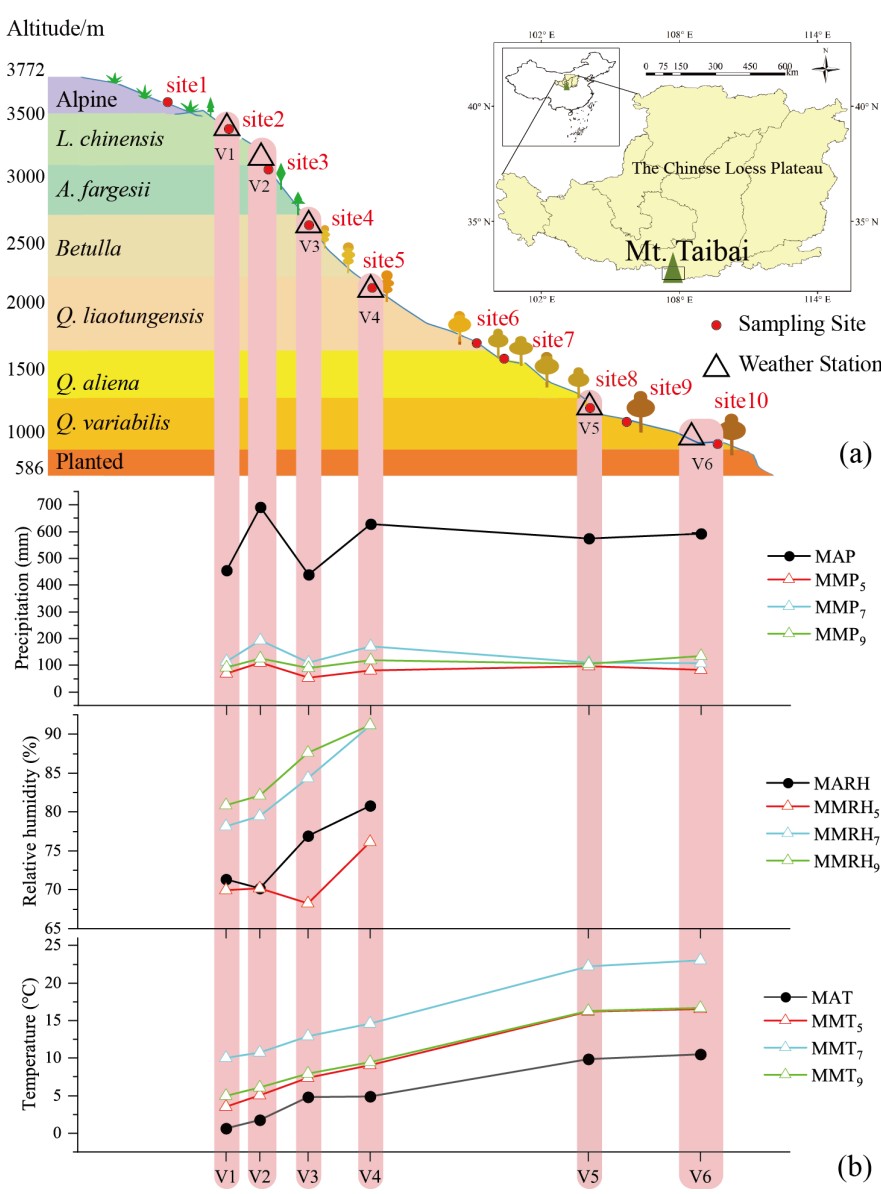


Figure-1








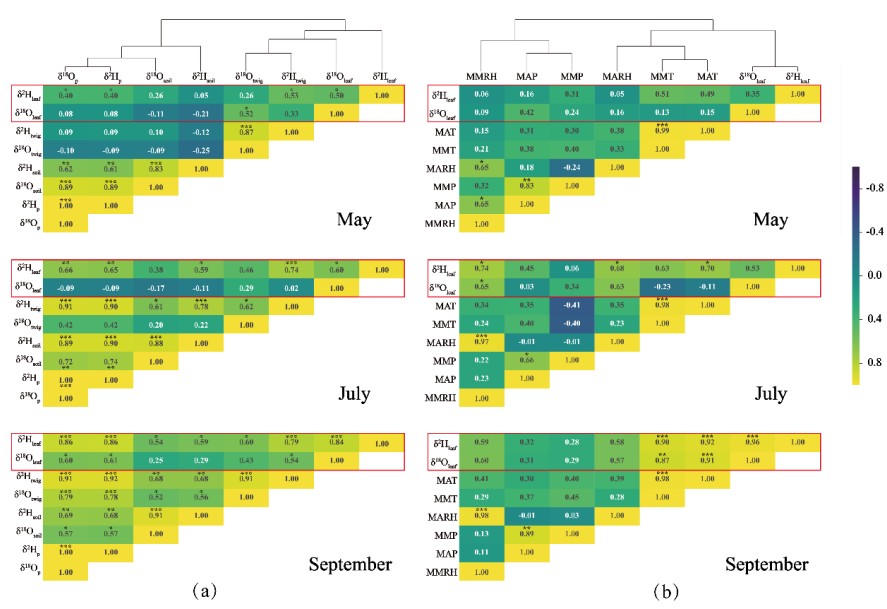

Figure-2





















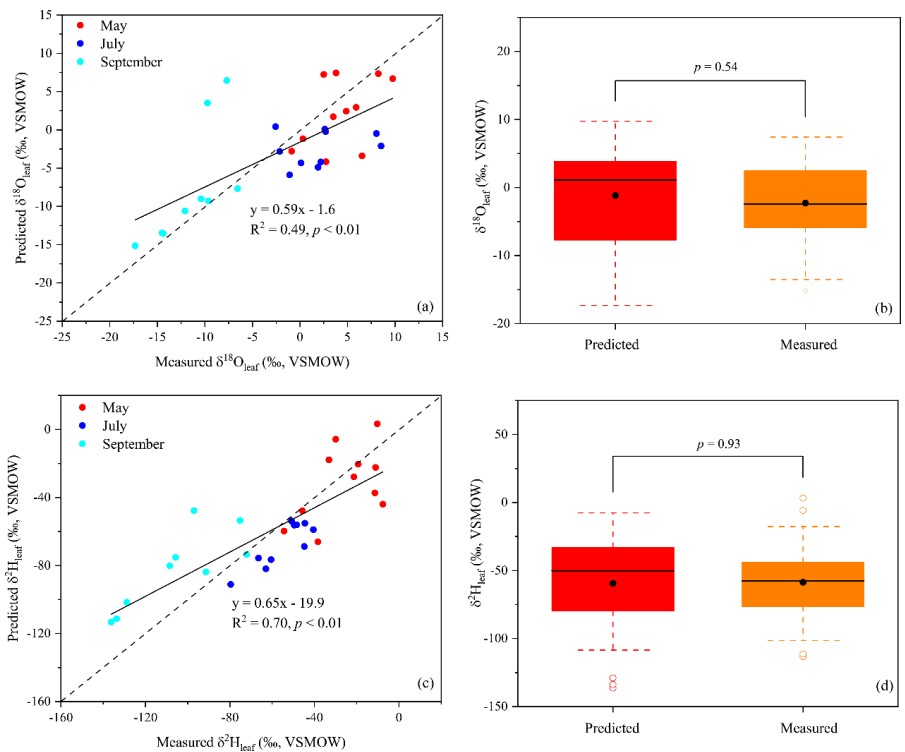


Figure-3



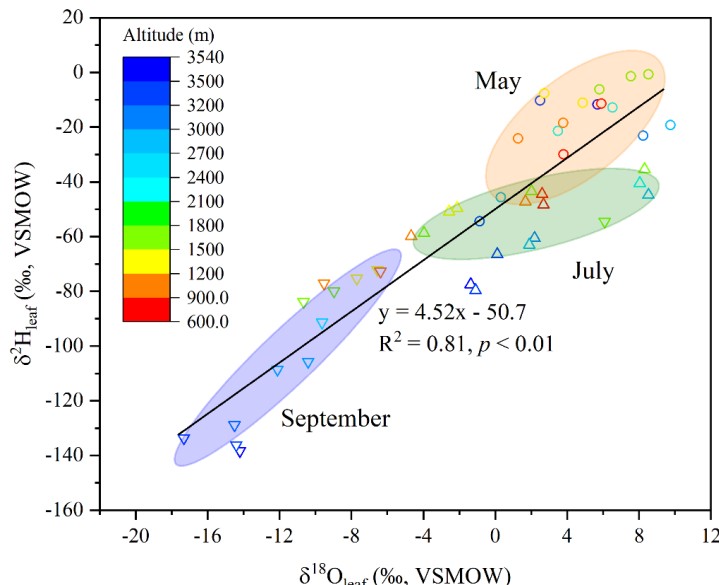


Figure-4

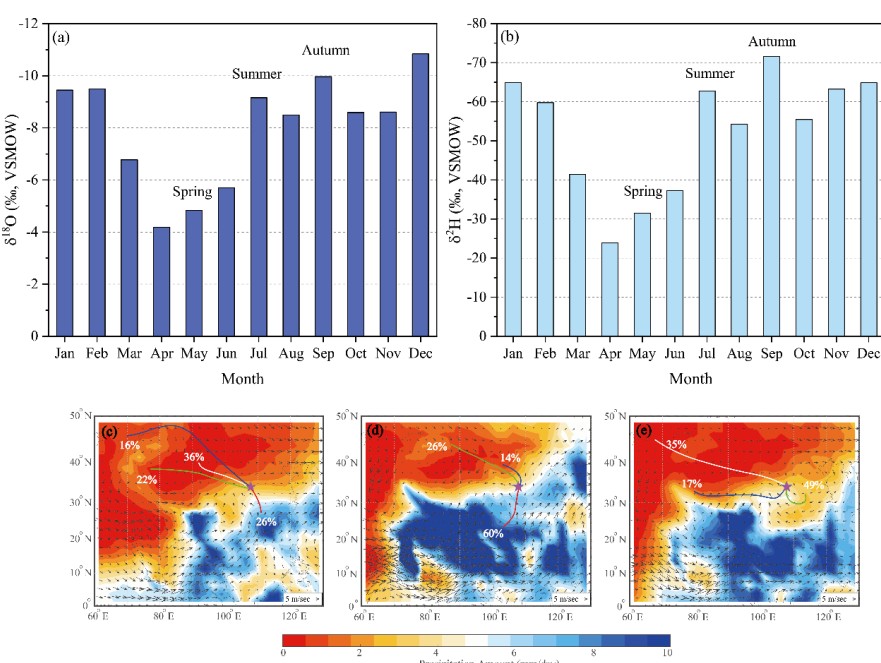


Figure-5





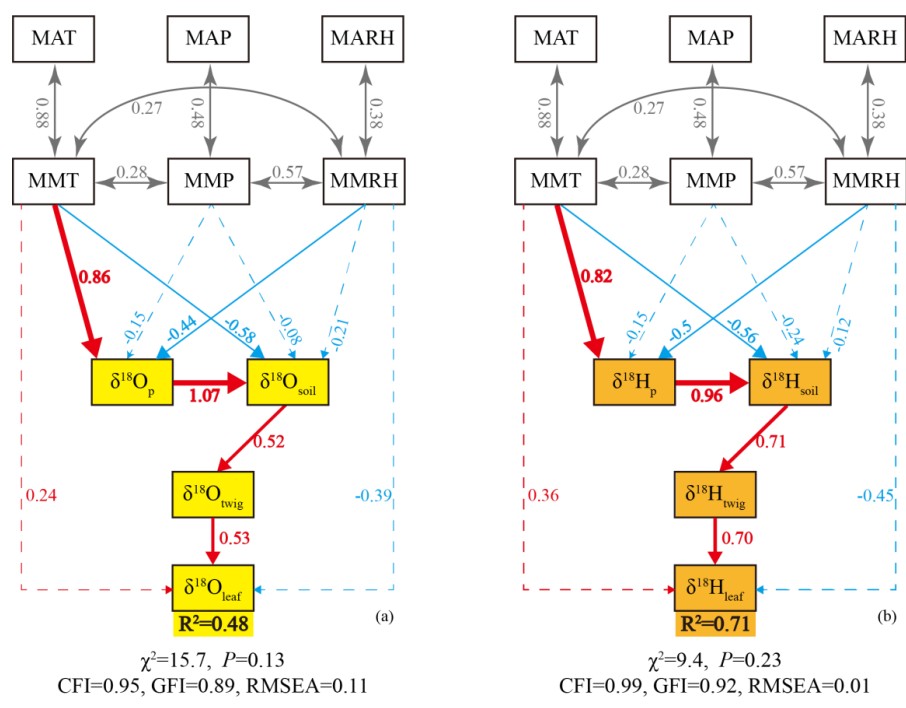


Figure-6

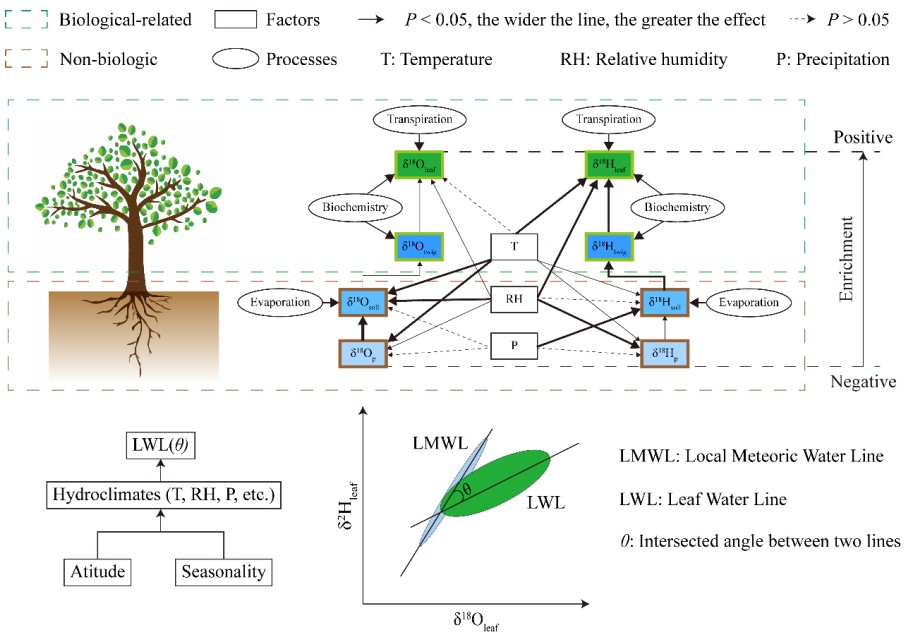


Figure-7