# Peer review of "Controls on leaf water hydrogen and oxygen isotopes: A local"

_Hydrology and Earth System Sciences, 2022_

## Referee Comment (RC1)

I appreciate the Editor to give me a chance to review the paper.

The manuscript "Controls on leaf water hydrogen and oxygen isotopes: A local investigation across seasons and altitude" presents a dataset on analysis of $\delta^{18}O_{leaf}$ and $\delta^2H_{leaf}$ together with isotopes from potential source waters and meteorological parameters along an elevation transect on the Chinese Loess Plateau. The research topic is important and within the scope of the journal.

But it seems a bit simple and not systematic in the content. The manuscript at present lacks novel results or theory that would provide a significant advance in this field.

1) The main conclusion of this paper has been confirmed by previous studies: the first-order control on $\delta18O_{leaf}$ and $\delta2H_{leaf}$ values was the source water,and the second-order control was the enrichment associated with biochemical and environmental factors(Cernusak et al., 2016; Barbour et al., 2017; Munksgaard et al., 2017). The experimental design and results of the paper are not innovative.

2) A large number of studies have shown that the enrichment associated with plant transpiration is an important factor affecting $\delta^{18}O_{leaf}$ and $\delta^2H_{leaf}$ values. However, the authors did not carry out research and discussion in this paper.

3)Plants and soils were sampled in May, July, and September 2020 (In the experimental design). Why only choose this three months? Is it persuasive?

4)Besides, what is the specific sampling interval?

5) Why only one or two deciduous and coniferous trees were chosen in each plot?

6) There are large differences in population and altitude between sampling points 5-8(Fig.1). But there is no weather station here.

7) In 4.1,these results argued with the recent global meta-analysis that $\delta^{18}O_{leaf}$ and $\delta^2H_{leaf}$ values reflect climatic parameters (i.e., RH and temperature) differently. What are the reasons for the controversial conclusion?

8) It seems a bit simple in the conclusion. It needs a stronger ending for the conclusion. Besides, it is suggested to supplement the existing deficiencies and prospects.

---

## Author Comment (AC1)

Comment:

*I appreciate the Editor to give me a chance to review the paper.*

*The manuscript "Controls on leaf water hydrogen and oxygen isotopes: A local investigation across seasons and altitude" presents a dataset on analysis of δ18Oleaf and δ2Hleaf together with isotopes from potential source waters and meteorological parameters along an elevation transect on the Chinese Loess Plateau. The research topic is important and within the scope of the journal. But it seems a bit simple and not systematic in the content. The manuscript at present lacks novel results or theory that would provide a significant advance in this field.*

Response:

Thanks a lot for your comments. Our study have two significant novel points: 1) the previous studies have always emphasized on the combined $\delta^{18}O$ and $\delta^2H$ values of leaf water ($\delta^{18}O_{leaf}$ and $\delta^2H_{leaf}$), few considering the respective responses or variations of $\delta^{18}O_{leaf}$ and $\delta^2H_{leaf}$. A recent global meta-analysis indicate that the respective $\delta^{18}O_{leaf}$ and $\delta^2H_{leaf}$ reflected differently, seen details in Cernusak et al. (2022; NP). However, our local-study supported that $\delta^2H_{leaf}$ responds more closely to xylem water than $\delta^{18}O_{leaf}$, but both $\delta^{18}O_{leaf}$ and $\delta^2H_{leaf}$ responds comparatively to climatic factors (RH, T), challenging the global meta-analysis (Cernusak et al., 2022); 2) We proposed a framework that control the leaf water isotope line by using multivariate statistical methods (Hierarchical clustering, Craig-Cordon model, Structural equation model, HYSPLIT, etc)

Reference

Cernusak, L. A., Barbeta, A., Bush, R., Eichstaedt R., Ferrio, J., Flanagan, L., Gessler, A., Martín-Gómez, P., Hirl, R., Kahmen, A., Keitel., C., Lai, C., Munksgaard, N., Nelson, D., Ogée J., Roden, J., Schnyder, H., Voelker, S., Wang L., Stuart-Williams, H., Wingate, L., Yu, W., Zhao, L., Cuntz, M., 2022. Do $^2H$ and $^{18}O$ in leaf water reflect environmental drivers differently? New Phytologist, DOI: 10.1111/nph.18113.

Comment:

*1) The main conclusion of this paper has been confirmed by previous studiesï¼ the first-order control on δ18O leaf and δ2Hleaf values was the source waterï¼and the second-order control was the enrichment associated with biochemical and environmental factorsï¼Cernusak et al., 2016; Barbour et al., 2017; Munksgaard et al., 2017). The experimental design and results of the paper are not innovative.*

Response:

Thanks. "The first-order control on $\delta^{18}O_{leaf}$ and $\delta^2H_{leaf}$ values was the source waterï¼and the second-order control was the enrichment associated with biochemical and environmental factorsï¼Cernusak et al., 2016; Barbour et al., 2017; Munksgaard et al., 2017)" is deed analyzed by previous studies, as discussed in Introduction section. Our studies analyzed the responses of respective ($\delta^{18}O_{leaf}$ and $\delta^2H_{leaf}$) and both to source waters (xylem water, soil water, and precipitation) and to meteorological parameters (temperature, RH). As above stated, we have two significant novel points.

Comment:

*2) A large number of studies have shown that the enrichment associated with plant transpiration*

*is an important factor affecting δ18Oleaf and δ2Hleaf values. However, the authors did not carry out research and discussion in this paper.*

Response:

Thanks. We have added more discussion on transpiration.

Comment:

*3)Plants and soils were sampled in May, July, and September 2020 (In the experimental design). Why only choose this three months? Is it persuasive?*

Response:

Thanks. The growing season lasts from late April to Early October on the Chinese Loess Plateau, so we selected the pre- (May), peak (July), and post-(September) growing season. Also, the precipitation $\delta^{18}O$ and $\delta^2H$ varies across months (Fig. 5a, b), which was caused by different moisture transport routes from HYSPLIT (Fig, 5c).

Additionally, we sampled at the same plots (ten plots) along an elevation transect from ~600 m to ~3600 m, the three repeated sampling is OK. If more, the sampling will be a burdensome work and the plants is not available for more repeated sampling.

Comment:

*4ï¼ Besides, what is the specific sampling interval?*

Response:

Thanks. The sampling plots were arranged for ten plots from ~600 m to ~3600 m along an elevation transect, which was detailed in Fig, 1 and supplementary Table S1.

The sampling plots were randomly selected in the first campaign from the bottom to top of mountain, then repeated by the next two sampling campaigns.

Comment:

*5) Why only one or two deciduous and coniferous trees were chosen in each plot?*

Response:

Thanks. There was a significant vegetation zone along an elevation transect of Mt. Taibai (Fig.1 and M&M), so we selected the dominant species at each zones.

Comment:

*6) There are large differences in population and altitude between sampling points 5-8(Fig.1). But there is no weather station here.*

Response:

Thanks. The weather stations along an elevation transect was very hard to settle up, the available weather stations were presented in Fig.1. We thank to Shaanxi Meteorological Bureau for supporting meteorological data. It is possible that more weather stations will be settled up along this elevation transects in the future.

Comment:

*7) In 4.1,these results argued with the recent global meta-analysis thatδ18Oleaf and δ2Hleaf values reflect climatic parameters (i.e., RH and temperature) differently. What are the reasons for the controversial conclusion?*

Response:

Thanks. It is really a good question, I think it is probably due to the scale difference, e.g., global vs. local. The reason needs to be further explored in the future.

Comment:

*8) It seems a bit simple in the conclusion. It needs a stronger ending for the conclusion. Besides, it is suggested to supplement the existing deficiencies and prospects.*

Response:

Thanks. We have strengthen the conclusion.

---

## Author Comment (AC2)

Comment:

*Liu et al, based on field sampling of leaf water and measuring isotope composition (δ18O and δ2H) along an elevation transect on the Chinese Loess Plateau belt to illustrate controls on leaf water hydrogen and oxygen isotopes. The results point that the first-order control on δ18O and δ2H was the source water, and the second-order control was the enrichment associated with biochemical and environmental factors. Overall, this is an important and hard-working investigation for deepening the understanding of the control of leaf water isotopic composition in field conditions. However, the current analysis is mediocre, the scientific questions are poorly elaborated, and new discoveries are lacking. Secondly, the results show that source water is the main control of leaf water isotopes, which is contrary to the previous results which indicate that relative humidity is the main control of leaf water isotopes both under leaf and ecosystem scale. Therefore, there requires more evidence to support your conclusion. I recommend major revisions before considering publication in this journal.*

Response:

Thanks a lot for your approvedness and suggestions. For the first question, we have two significant novel points: 1) the previous studies have always emphasized on the combined $\delta^{18}O$ and $\delta^{2}H$ values of leaf water ($\delta^{18}O_{leaf}$ and $\delta^{2}H_{leaf}$), few considering the respective responses or variations of $\delta^{18}O_{leaf}$ and $\delta^{2}H_{leaf}$. A recent global meta-analysis indicate that the respective $\delta^{18}O_{leaf}$ and $\delta^{2}H_{leaf}$ reflected differently, seen details in Cernusak et al. (2022; NP). However, our local-study supported that $\delta^{2}H_{leaf}$ responds more closely to xylem water than $\delta^{18}O_{leaf}$, but both $\delta^{18}O_{leaf}$ and $\delta^{2}H_{leaf}$ responds comparatively to climatic factors (RH, T), challenging the global meta-analysis (Cernusak et al., 2022); 2) We proposed a framework that control the leaf water isotope line by using multivariate statistical methods (Hierarchical clustering, Craig-Cordon model, Structural equation model, HYSPLIT, etc). The leaf water isotope line was generated by our analysis, which provides an important baseline for leaf-derived organic matter such as cellulose and leaf wax.

For the second question, our results were actually consistent with the previous conclusion. We proposed a hierarchical control on leaf water isotopes, as the following figure. The first-order control is source water, which is also affected by climate factors (e.g., T, RH). The climatic factors (e.g. RH) affect source water and directly affect leaf water isotopes. Without considering source water, the RH is the main control on leaf water isotopes.

[Figure]

Comment:

***Minor revision:***

*Line 148-150,  "For the plants, one or two deciduous and coniferous trees were chosen in each plot, and several large leaves and suberized twigs were collected for each species." Here should be describe in details.*

Response:

Thanks. We have added more details.

Comment:

*Line 214-215, How was the   $\delta^{?}$   and $\delta^{?}$ obtained? Here should be describe in details.*

Response:

Thanks. We have added new citation, the kinetic fractionation equations for hydrogen and oxygen isotopes can be found in the reference.

Comment:

*Line 264, predicted δ18O and δ2H values of leaf water were quiet simper, and need consider non stead state (NSS) under complex environment condition.*

Response:

Thanks. It is a good question. We used the steady-state condition in this study because our sampling campaigns take place during the day when leaf water is generally near isotopic steady state because chloroplasts are mostly located near to the evaporative sites (Cernusak et al., 2016). The non-steady state effects on leaf water isotopes were expected at night because of low stomatal conductance (Cernusak et al., 2005; Cuntz et al., 2002; Cernusak et al., 2016).

Comment:

*Line 456-459,there need more deep-seated analysis but not common knowledge in this area.*

Response:

Thanks. We have depleted it. This needs to be further explored in the future.

Comment:

*Line 466-469, The conclusion is too simple and needs to be further explored.*

Response:

Thanks. We have strengthen the conclusion.

---

## Author Response (AR2)

Dear Editor,

Thank you for your letter and for the reviewer's comments concerning our manuscript entitled "Controls on leaf water hydrogen and oxygen isotopes: A local investigation across seasons and altitude" (hess-2022-246R1). Those comments are all valuable and very helpful for improving our paper and for future studies. We have studied those comments carefully and have made corrections or explanations accordingly. Responses to the referee's comments are as follows:

**Comments to the author:**

*The reviewers are generally satisfied with the revision. But both reviewers commented on specific areas that deserve further changes and in-depth analyses. Reviewers also pointed out that the language quality of this manuscript requires significant improvements. I concur with the reviewers' assessment and ask the authors to make a revision again with the reviewers' comments in mind. Thank you!*

Thank you. We read the reviewers' comments and carefully considered them, and we have revised the paper accordingly and answered the questions carefully.

We also re-refresh and polish the language.

**Reviewer #1:**

Comment:

*The results argued with the recent global meta-analysis that δ18Oleaf and δ2Hleaf values reflect climatic parameters (i.e., RH and temperature) differently(Line 313-315). I suggest adding a detailed discussion and explaining the possible reasons.*

Response:

Thank you. We have added more discussion and possible explanations for this (Lines 318-325). Our local investigation was consistent with the global meta-analysis of different responses of $\delta^{18}O_{leaf}$ and $\delta^2H_{leaf}$ to the isotopic composition of the source water and meteorological conditions. However, we observed the equivalent responses of $\delta^{18}O_{leaf}$ and $\delta^2H_{leaf}$ to climatic factors, which is likely due to the difference of study scales.

Comment:

*The language of this paper needs to be further improved.*

Response:

Thank you.   We have polished the language.

Comment:

*Reviewer#1 also asked for tracked changes in future revisions.*

Response:

Thank you.   We made the revision in the tracked change of the marked-up version.

**Reviewer #2:**

Comment:

*♣ The author has made revisions to the article, responding to the questions raised, and basically maintaining the general situation of the previous version. In generally. the explanation for the*

*results was reasonable. However, there are still a few issues that need to be addressed before I would recommend it for publication by HESS.*

Response:

Thank you for your positive comments on this manuscript. We have addressed all comments according to two reviewers' constructive comments.

Comment:

♣ *There is a good one-to-one correspondence between the leaf water and the stem water isotope data set, but it is not possible to have a similar one-to-one correspondence between the thermometer humidity to calculate the correlation between them. Especially in mountainous areas, the heterogeneity of temperature and humidity is more intense. What is the impact of the results on this situation?*

Response:

Thank you. We agree with your valuable comments on the heterogeneous difference of temperature and humidity in the mountainous regions. Our results also showed the distinct difference of temperature, humidity, and stem and leaf water isotopes. The details were seen as below:

[Figure]

Comment:

*♣ What is the time interval between each sample and the last rainfall? Could it be related to the fact that the isotopic signal of leaf water in September was extremely poor?*

Response:

Thank you. We thank the Shaanxi Meteorological Bureau for supporting meteorological data (precipitation, temperature, and RH, etc.) along an elevation transect. The precipitation amount was measured after each precipitation event at the station sites.

In this study, we focused on the effects of mean annual precipitation (MAP) and mean monthly precipitation (MMP) on leaf water isotopes ($\delta^{18}O_{leaf}$ and $\delta^2H_{leaf}$ values), so we did not consider the time interval between three sampling and the last rainfall. The comment is good. Maybe we will conduct a detailed pinpoint measurement over a short period (several days) in the future.

In our study area, we guess that the relatively poor isotope signals of leaf water in September are because most of the species have been dried or died in September because of the mountainous climate (Zhao et al., 2018; Liu, 2021).

References

Zhao M., Wang Y., Xue F., Zuo W., Xing K., Wang G., Kang M., Jiang Y., 2018. Elevational patterns and ecological determinants of mean family age of angiosperm assemblages in temperate forests within Mount Taibai, China. *Journal of Plant Ecology*, 11, 919-927.

Liu, J. 2021. Seasonality of the altitude effect on leaf wax n-alkane distributions, hydrogen and carbon isotopes along an arid transect in the Qinling Mountains. *Science of the Total Environment*, 778, 146272.

Comment:

*♣ How to consider the effects of biodiversity on leaf water sampling representation? Especially the alpine meadow ecosystem?*

Response:

Thank you. A significantly vertical vegetation zones existed (Fig. 1), so we collected the dominant species for representatives at each zone across the elevation transect (Tang et al., 2006).

Actually at the alpine (> 3350 m asl), the species is monotonous (e.g., *Rhododendron sp.*) (Zhao et al., 2018; Tang et al., 2006).

References

Tang Z., Fang J., 2006. Temperature variation along the northern and southern slopes of Mt. Taibai, China. *Agricultural and Forest Meteorology*, 139, 200–207.

Zhao M., Wang Y., Xue F., Zuo W., Xing K., Wang G., Kang M., Jiang Y., 2018. Elevational patterns and ecological determinants of mean family age of angiosperm assemblages in temperate forests within Mount Taibai, China. *Journal of Plant Ecology*, 11, 919-927.

Comment:

*♣ The current conclusions about the control of water isotope signals by source water on leaves have strong regional and atmospheric circulation characteristics and should be highlighted for the current conclusion.*

Response:

Thank you. We have highlighted the regional characteristics in the text.

---

## Author Response (AR3)

Dear Editor,

Thank you for your letter and for the reviewer's comments concerning our manuscript entitled "Controls on leaf water hydrogen and oxygen isotopes: A local investigation across seasons and altitude" (hess-2022-246). Responses to the referee's comments are as follows:

**Comments to the author:**

*1. Please note that any change to the list of authors at this stage of the review needs to be approved by the editor. For a subsequent revision, please indicate in the "Author's response" the reasons for adding to the list of the author the co-author Ying Zhao. If necessary, please check and adjust the "Author contribution" section of the \*.pdf manuscript file accordingly.*

Thank you. The co-author Ying Zhao was actually listed in earlier versions (the first and second versions), but was removed as some internal reasons. Ultimately, we decide to keep unchanged for the coauthor list. Please understand.

On the other hand, we adjusted the location of fundings, and also added a relevant reference of our recent

published paper, i.e., Liu, J., Jiang, C., Guo, L., Hu, J: Ecohydrological separation in a pair catchments

covered with natural grassland and planted forestland on the Chinese Loess Plateau: Evidence from a

one-year stable isotope observation. Hydrol. Process. 36, e14778, 2022.

*2. Coloured or marked text in \*.pdf manuscript file is not allowed. Please provide a clean version of \*pdf manuscript file (with black text) with the next revision.*

Thank you. We have provided a clean version of \*pdf manuscript file (with black text).